# The MOV10 RNA helicase is a dosage-dependent host restriction factor for LINE1 retrotransposition in mice

Yongjuan Guan[1,2], Hongyan Gao[3], N. Adrian Leu[1], Anastassios Vourekas[4,5], Panagiotis Alexiou[4], Manolis Maragkakis[4], Zhenlong Kang[1,6], Zissimos Mourelatos[4], Guanxiang Liang[3], P. Jeremy Wang[1]*

**1** Department of Biomedical Sciences, University of Pennsylvania School of Veterinary Medicine, Philadelphia, Pennsylvania, United States of America, **2** College of Life Sciences, Capital Normal University, Beijing, China, **3** Center for Infectious Disease Research, School of Medicine, Tsinghua University, Beijing, China, **4** Department of Pathology and Laboratory Medicine, Perelman School of Medicine, University of Pennsylvania, Philadelphia, Pennsylvania, United States of America, **5** Department of Biological Sciences, Louisiana State University, Baton Rouge, Louisiana, United States of America, **6** State Key Laboratory of Reproductive Medicine, Nanjing Medical University, Nanjing, China

* pwang@vet.upenn.edu

**Data Availability Statement:** The CLIP-seq and RNA-seq data that support the findings of this study are publicly available from NCBI under the GEO accession no: GSE217336.

## Abstract

Transposable elements constitute nearly half of the mammalian genome and play important roles in genome evolution. While a multitude of both transcriptional and post-transcriptional mechanisms exist to silence transposable elements, control of transposition *in vivo* remains poorly understood. MOV10, an RNA helicase, is an inhibitor of mobilization of retrotransposons and retroviruses in cell culture assays. Here we report that MOV10 restricts LINE1 retrotransposition in mice. Although MOV10 is broadly expressed, its loss causes only incomplete penetrance of embryonic lethality, and the surviving MOV10-deficient mice are healthy and fertile. Biochemically, MOV10 forms a complex with UPF1, a key component of the nonsense-mediated mRNA decay pathway, and primarily binds to the 3′ UTR of somatically expressed transcripts in testis. Consequently, loss of MOV10 results in an altered transcriptome in testis. Analyses using a LINE1 reporter transgene reveal that loss of MOV10 leads to increased LINE1 retrotransposition in somatic and reproductive tissues from both embryos and adult mice. Moreover, the degree of LINE1 retrotransposition inhibition is dependent on the *Mov10* gene dosage. Furthermore, MOV10 deficiency reduces reproductive fitness over successive generations. Our findings demonstrate that MOV10 attenuates LINE1 retrotransposition in a dosage-dependent manner in mice.

## Author summary

Transposable elements (TEs), including L1 and SINEs, are abundant in the genome and play important roles in evolution, development, and diseases. While TEs propagate in individuals and across generations, the host organism needs to suppress them, resulting in an ongoing arms race between TEs and the host genome. L1, a retrotransposon, accounts

**Funding:** Funding support was received from National Institute of Child Health and Human Development grants R01 HD069592 and P50 HD068157 (PJW) and National Institute of General Medical Sciences R01 GM123512 (ZM). The funders had no role in study design, data collection and analysis, decision to publish, or preparation of the manuscript.

**Competing interests:** The authors have declared that no competing interests exist.

for about 17% of the mammalian genome. L1 encodes two proteins, which bind to the L1 transcript to form L1 ribonucleoprotein particles. L1 proliferates in the genome via retrotransposition. A multitude of transcriptional and post-transcriptional mechanisms exist to suppress TEs, however, retrotransposition of TEs remains poorly understood. L1 ribonucleoprotein particles are associated with a large number of host proteins, one of which is the MOV10 RNA helicase. MOV10 exhibits anti-viral activities against retroviruses such as HIV-1. In cultured cells, MOV10 is an inhibitor of retrotransposition of L1, SINEs, and IAP. Although MOV10 is expressed in a broad range of tissues, loss of MOV10 causes only incomplete penetrance of embryonic lethality. The viable MOV10-deficient mice are grossly normal and fertile. Importantly, analyses using a L1 transgene reporter reveal that MOV10 inhibits L1 retrotransposition in both somatic tissues and reproductive tissues in a gene dosage-dependent manner. Therefore, MOV10 functions as a host restriction factor for L1 and possibly other transposable elements *in vivo*.

## Introduction

Transposable elements (TEs) constitute ~40% of the mammalian genome. Despite sometimes being called "junk" DNA, TEs play important roles in genome evolution, development, and diseases [1–4]. TEs have an enormous capacity to amplify in the host genome. On one hand, integration of TEs into new genomic sites can change the level or pattern of neighboring gene expression or generate new genes, resulting in greater genetic diversity that could be beneficial to the host [2,3]. On the other hand, genomic insertion can disrupt gene function and cause immediate harm to the host cell/organism. Indeed, many sporadic genetic diseases in animal species and humans are caused by transposon insertion [1]. While TEs exploit the host cellular machinery to propagate, the host in turn has evolved multiple mechanisms to suppress their mobilization to protect genome integrity [5]. The outcome of the ongoing arms race between the parasitic TEs and the host is particularly critical in the germline, where TE-induced genetic changes can impact fertility and the genetic integrity over subsequent generations.

Retrotransposons, including endogenous retroviruses, are the only known active TEs in mouse and human. While the vast majority of retrotransposons are truncated inactive copies, a subset of elements remains intact and are capable of transposition–the insertion of new copies at new genomic locations. LINE1 (long interspersed nuclear element-1; L1), SINEs (short interspersed nuclear elements), and LTR (Long terminal repeat) retrotransposons are active in mouse, while L1s, SINEs, and SVAs are active in human. L1 is the most abundant class of TEs in mammals, accounting for about 17% of the genome in mouse and human. It is estimated that up to 3000 copies of L1 in mouse [6] and ~100 copies of L1 in human are intact and active [7,8].

L1 retrotransposons mobilize in the genome through a "copy and paste" mechanism using reverse transcription. The 6-kb full-length L1 element contains two open reading frames (ORF1 and ORF2). The L1 ORF1 protein is an RNA-binding protein that forms a trimer and possesses nucleic acid chaperone activity [9]. The ORF2 protein exhibits endonuclease and reverse transcriptase activities [10,11]. ORF1 and ORF2 proteins bind the L1 mRNA transcript to form L1 ribonucleoprotein particles that are imported into the nucleus, resulting in possible integrations at new genomic locations. SINEs do not encode proteins and rely on L1-encoded ORF2 protein for retrotransposition [12,13]. Reverse transcription of L1 and SINE RNAs is primed by ORF2-nicked DNA in a process called target primed reverse transcription. Subsequently host-encoded DNA repair enzymes complete the integration reaction.

The host has evolved multiple mechanisms to inhibit retrotransposons to protect genome integrity, including DNA methylation [14,15], histone modifications [16,17], and small-non-coding RNAs such as piRNAs [18–25]. L1 contains a 5′ untranslated region (5′ UTR) that functions as its promoter. Normally, L1 is methylated at the CpG dinucleotides in its promoter and thus transcriptionally silenced. We previously identified MOV10L1, an RNA helicase specifically expressed in mouse germ cells, as a critical regulator of retrotransposons [26,27]. MOV10L1 interacts with all three mouse Piwi proteins and is required for biogenesis of all piRNAs [27–29]. Mechanistically, MOV10L1 binds directly to piRNA precursors to initiate piRNA biogenesis and its RNA helicase activity is required for this process [30–32]. piRNAs post-transcriptionally degrade L1 transcripts through Piwi proteins and transcriptionally silences active L1 transposons through de novo methylation of L1 promoters in the germline [33]. Inactivation of piRNA biogenesis factors such as MOV10L1 and Piwi proteins leads to de-silencing of retrotransposons in germ cells, meiotic arrest, and male infertility in mice.

Genetic studies have identified a large number of genes that are important for retrotransposon silencing in the germline [5]. Most of these genes function in the piRNA and DNA methylation pathways. These factors suppress transcription of retrotransposons and/or cause post-transcriptional degradation of retrotransposon transcripts. However, upregulation of retrotransposon transcripts does not necessarily lead to a proportional increase in new retrotransposition, suggesting that additional host factors block retrotransposition. Notably, proteomic studies have identified several dozen of cellular proteins that are associated with L1 ribonucleoprotein particles [34–36], one of which is MOV10 (Moloney leukemia virus 10), a homologue of MOV10L1. While MOV10L1 is germ cell-specific, MOV10 is widely expressed. MOV10 is also an RNA helicase [37]. MOV10 plays a role in host defense against retroviruses and inhibits the activity of retroviruses such as HIV-1 at multiple stages [38–41]. Notably, MOV10 is known to inhibit retrotransposition of L1, SINE, and IAP retrotransposons in cell culture-based assays [42–45]. MOV10 interacts with RNASEH2, a nuclear ribonuclease, which hydrolyzes the RNA strands of L1-specific RNA-DNA hybrids in a MOV10-dependent manner [46]. MOV10 cooperates with two terminal uridyltransferases, TUT4 and TUT7, to promote uridylation of human L1 mRNA to destabilize it and inhibit its reverse transcription [47]. While these studies have provided important insights into the role of MOV10 in preventing L1 retrotransposition in cultured cells, such a role for MOV10 in intact host organisms remains poorly understood. In a previous study [48], loss of MOV10 results in complete embryonic lethality in mice, and MOV10 suppresses retroelements and regulates brain development. Here we find that loss of MOV10 causes partial embryonic lethality. Using a L1 reporter transgene, we demonstrate that MOV10 is a host restriction factor for L1 retrotransposition in both somatic cells and germ cells in mice.

## Results

### Loss of MOV10 causes embryonic lethality with incomplete penetrance

In adult mice, the MOV10 protein is highly expressed in several tissues including testis, ovary, kidney, spleen, and liver, present at a low level in lung, but not detected in brain, heart, and muscle (Fig 1A). In HEK293T cells, MOV10 is associated with UPF1, a key component of the nonsense-mediated mRNA decay pathway [37,49,50]. Similar to MOV10, UPF1 is also abundantly expressed in multiple adult tissues but not heart and muscle (Fig 1A).

To identify MOV10-associated proteins in tissues, we performed immunoprecipitation of mouse testicular extracts with anti-MOV10 antibody. Both the putative MOV10 band and one extra band in the immunoprecipitated proteins were subjected to mass spectrometry for protein identification (S1A Fig). In addition to the identification of MOV10, one associated

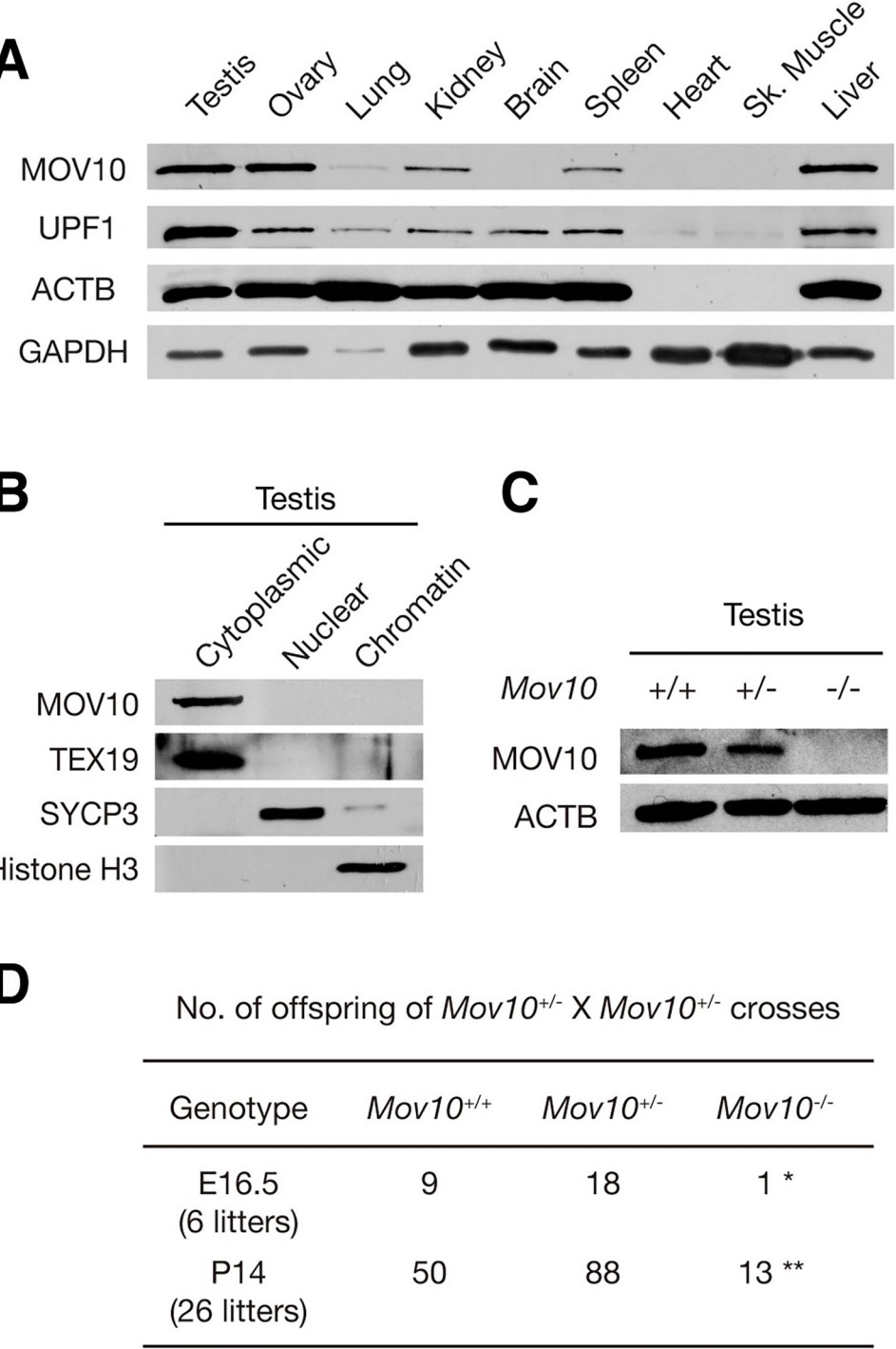

**Fig 1. Loss of MOV10 leads to incomplete penetrance of embryonic lethality.** (A) The expression of MOV10 in adult (8-week-old) mouse tissues. UPF1 is a MOV10-interacting protein (S1 Fig) [37]. ACTB and GAPDH serve as loading controls. Note that heart and skeletal muscle contain little ACTB. (B) MOV10 is cytoplasmic in 8-week-old testes. TEX19, SYCP3, and histone H3 serve as cytoplasmic, nuclear, and chromatin controls respectively [67]. (C) Western blot analysis of MOV10 in testes from 8-week-old $Mov10^{+/+}$, $Mov10^{+/-}$, and $Mov10^{-/-}$ mice. (D) Number of offspring from $Mov10^{+/-} \times Mov10^{+/-}$ crosses at E16.5 and postnatal day 14 (P14).

protein was identified as UPF1 (S1A Fig). The association of MOV10 and UPF1 in testis was confirmed by co-immunoprecipitations and Western blot analyses (S1B and S1C Fig), showing that, as in human HEK293T cells, MOV10 forms a complex with UPF1 in mouse tissues.

Because of the high abundance of MOV10 in testis, we examined its expression in developing testes and found that the MOV10 abundance was constant in testes from postnatal day 6 (P6) to P56 (adulthood) (S2A Fig). Western blot analysis using fractionated testicular extracts showed that MOV10 was cytoplasmic (Fig 1B). Immunofluorescence of MOV10 in frozen sections from embryonic 16.5 (E16.5), P0, and 8-week-old testes showed that MOV10 was highly expressed in the cytoplasm of Sertoli cells, gonocytes, spermatogonia, leptotene and zygotene spermatocytes, but present at a low level in pachytene spermatocytes and not detected in spermatids (S2B Fig). In the adult ovary, MOV10 was abundant in granulosa cells but not in oocytes (S2B Fig). These results demonstrate that MOV10 is a broadly expressed cytoplasmic protein in testis and ovary.

As its relatively ubiquitous expression suggests that conventional knockout of *Mov10* could be embryonic lethal, we generated a *Mov10* conditional (floxed) allele (referred to as *Mov10*$^{fl}$) through gene targeting in embryonic stem (ES) cells by homologous recombination (S3 Fig). MOV10 has two RNA helicase domains at its C-terminal half: the DEXXQ/H-box helicase domain of Helz-like helicases and the C-terminal helicase domain of UPF1-like family helicases (S3A Fig). Cre-mediated deletion of floxed exons 8–14 deletes the first RNA helicase domain and causes a frame shift in the mutant transcript (S3B Fig). We used *Ddx4*-Cre and *Amh*-Cre to inactivate *Mov10* in germ cells and gonadal somatic cells respectively. *Ddx4*-Cre (also called Vasa-Cre) expression begins in germ cells at embryonic day 15 in both sexes [51]. *Amh* (anti-Mullerian hormone)-Cre is specifically expressed in Sertoli cells in testis and granulosa cells in ovary [52]. Immunofluorescence analysis revealed that, as expected, MOV10 was absent in germ cells but present in Sertoli cells in *Mov10*$^{fl/-}$ *Ddx4*-Cre testis, whereas MOV10 showed the opposite expression pattern in *Mov10*$^{fl/fl}$ *Amh*-Cre testis (S3C Fig). However, to our surprise, both *Mov10*$^{fl/-}$ *Ddx4*-Cre and *Mov10*$^{fl/fl}$ *Amh*-Cre conditional knockout mice were fertile.

We next used the ubiquitously expressed *Actb*-Cre to generate *Mov10* global knockout mice. *Actb*-Cre is under the control of the strong β-actin promoter [53]. Strikingly, intercross of *Mov10*$^{+/-}$ mice produced viable *Mov10*$^{-/-}$ (global knockout) offspring but at a reduced frequency, suggesting that *Mov10* deficiency causes partial embryonic lethality (Fig 1D). At E16.5, only one live *Mov10*$^{-/-}$ embryo was observed, confirming the partial lethality phenotype (Fig 1D). Our *Mov10* mutant mice are on a mixed genetic background (129 x C57BL/6 x FVB), possibly influencing penetrance. The global knockout (*Mov10*$^{-/-}$) mice were used in all the subsequent experiments. Histology of *Mov10*$^{-/-}$ testis and ovary did not reveal obvious defects (S4 Fig). Western blot analysis showed that the MOV10 protein was reduced in *Mov10*$^{+/-}$ testis and absent in *Mov10*$^{-/-}$ testis (Fig 1C). In addition, MOV10 was absent in the *Mov10*$^{-/-}$ testis by immunofluorescence, demonstrating that the knockout allele is null (S3C Fig). Strikingly, the surviving *Mov10*$^{-/-}$ mice were grossly normal and fertile. These results demonstrate that MOV10 is important for embryogenesis but that the penetrance of lethality caused by loss of MOV10 is incomplete.

## MOV10 binds preferentially to somatic transcripts and 3′ UTRs in testis

Because MOV10 is an RNA-binding protein [37], we sought to identify its RNA targets in wild type P21 testis. We performed high-throughput sequencing after cross-linking and immuno-precipitation (HITS-CLIP or CLIP-seq) (Fig 2A) [54]. Autoradiography after CLIP revealed specific MOV10-RNA protein complexes in the testis (Fig 2B). We extracted RNA from the

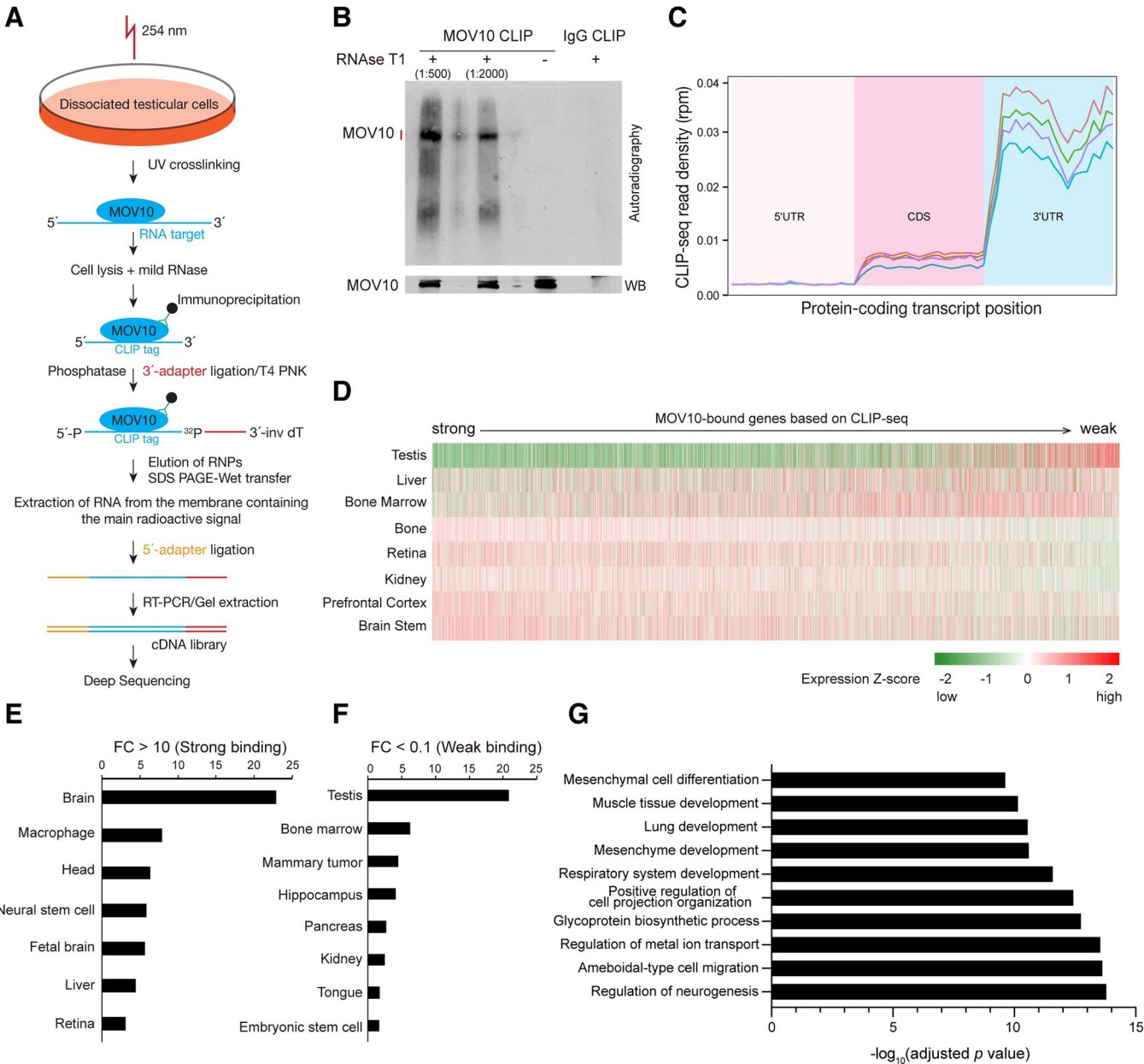

**Fig 2. Transcriptome-wide identification of MOV10 RNA targets in testis by HITS-CLIP.** (A) Graphic overview of the HITS-CLIP experimental strategy. (B) Autoradiography and Western blot analysis of the MOV10-RNA complexes from CLIP in testicular cells. Rabbit IgG serves as a negative control. Four libraries were constructed from RNA extracted from the main radioactive signal (red vertical line) and the higher position. (C) Distribution of MOV10 CLIP tags along protein-coding mRNAs (5' UTR, coding sequence, and 3' UTR). Y-axis, reads per million (RPM). (D) Expression heatmap of MOV10-bound transcripts in mouse testis and somatic tissues from 3-week-old mice (S5 Table). The transcripts were ordered from strong to weak based on MOV10-binding strength from the CLIP data. (E) Relative abundance of MOV10 strongly bound transcripts in different tissues. (F) Relative abundance of MOV10 weakly bound transcripts in different tissues. (G) Enrichment of GO categories in MOV10 highly-bound transcripts. A selection of top GO categories was plotted (S1 Table).

main radioactive signal band and constructed cDNA libraries for sequencing. The fold enrichment in CLIP-seq data was calculated by normalization using the P21 testis RNA-seq data. Our analysis showed that 2305 transcripts were highly bound by MOV10 (fold enrichment > 10), and 7174 transcripts moderately bound by MOV10 (1 < fold

enrichment < 10, S1 Table). Quantifying the distribution of MOV10 CLIP tags showed that they were highly enriched in the 3′ UTRs relative to transcript coding regions (Fig 2C). However, we noticed an inverse relationship between MOV10-binding and transcript expression level in testis (Fig 2D), with highly bound transcripts expressed at low levels in testis, while those that were weakly bound by MOV10 were more highly expressed. MOV10 interacts with UPF1 in testis (S1 Fig). CLIP-seq in mouse ES cells identified 210 UPF1-bound RNA targets [49]. 41 of these UPF1 targets (20%) overlapped with MOV10 highly bound CLIP targets (fold enrichment > 10), suggesting that they share RNA targets (S1 Table).

We next examined the expression of MOV10-bound transcripts in mouse somatic tissues by re-analyzing the RNA-seq data from these tissues (Fig 2D). We found that the transcripts highly bound by MOV10 in testis were highly expressed in somatic tissues, particularly in brain (Fig 2D and 2E), while those weakly bound by MOV10 in testis were highly expressed in testis (Fig 2D and 2F). GO analysis of MOV10 highly bound transcripts revealed enrichment of functional categories in regulation of neurogenesis, ameboidal-type cell migration, respiratory system development, and muscle tissue development (Fig 2G). These results raised the intriguing possibility that MOV10 modulates the testis transcriptome by binding to somatically expressed transcripts in the testis.

## Altered transcriptome in *Mov10*-deficient testis

To determine the effect of loss of MOV10 on the testicular transcriptome, we performed RNA-seq analysis of testes from P21 *Mov10*$^{+/+}$ and *Mov10*$^{-/-}$ mice. Using the cutoff (Fold change>2, FDR<0.05, RPM>1 in at least half of libraries), we found 926 down-regulated protein-coding genes and 60 upregulated protein-coding genes in *Mov10*$^{-/-}$ testes (Fig 3A and S2 Table). Notably, the upregulated genes in *Mov10*$^{-/-}$ testes were highly expressed in somatic tissues, such as retina, hippocampus, and liver (Fig 3B). However, the downregulated genes in *Mov10*$^{-/-}$ testes were strongly expressed in the testis in comparison with somatic tissues (Fig 3C). Consistently, GO analysis revealed that the down-regulated genes were highly enriched in several reproductive processes such as spermatid differentiation, fertilization, and sperm motility (Fig 3D), whereas the up-regulated genes were moderately enriched in one GO term: negative regulation of viral process (the adjusted $p$ value < 0.003; S2 Table). We performed correlation analysis between MOV10-binding (CLIP-seq) and the differential gene expression in *Mov10*$^{-/-}$ testis but did not find a correlation. MOV10 was reported to decrease the level of L1 transcripts in somatic cell lines [43,45]. Therefore, we examined whether L1 transcripts were affected in *Mov10*-deficient mouse testis. Our analysis showed that the overall abundance of L1 transcripts was not changed in *Mov10*-deficient testis in comparison with wild type controls (Fig 3E). Collectively, these results support that MOV10 modulates the transcriptome in testis.

## MOV10 inhibits L1 retrotransposition in mice

MOV10 inhibits retrotransposition of L1, SINE, and IAP retrotransposons in cell culture-based retrotransposition assays [42,43,45,46,48], suggesting that it may act as a host restriction factor for retrotransposons *in vivo*. As endogenous L1 elements are highly repetitive, it is difficult to distinguish new L1s from preexisting L1s. Therefore, we used an L1 reporter transgene, referred to as L1$^{tg}$ [55]. This L1 transgene has several key features: 1) its transcription is under the control of an endogenous mouse L1 promoter, which is methylated and thus repressed *in vivo*; 2) its two ORFs are codon-optimized to maximize translation; 3) it contains an intron in the disrupted GFP cassette in its 3′ UTR that serves as an indicator of retrotransposition; and 4) the transgene is single copy (inserted in the first intron of the *Tnr1* gene on Chr. 1) (Fig 4A) [55]. As the intron is expected to be spliced out following transcription, new L1 insertions

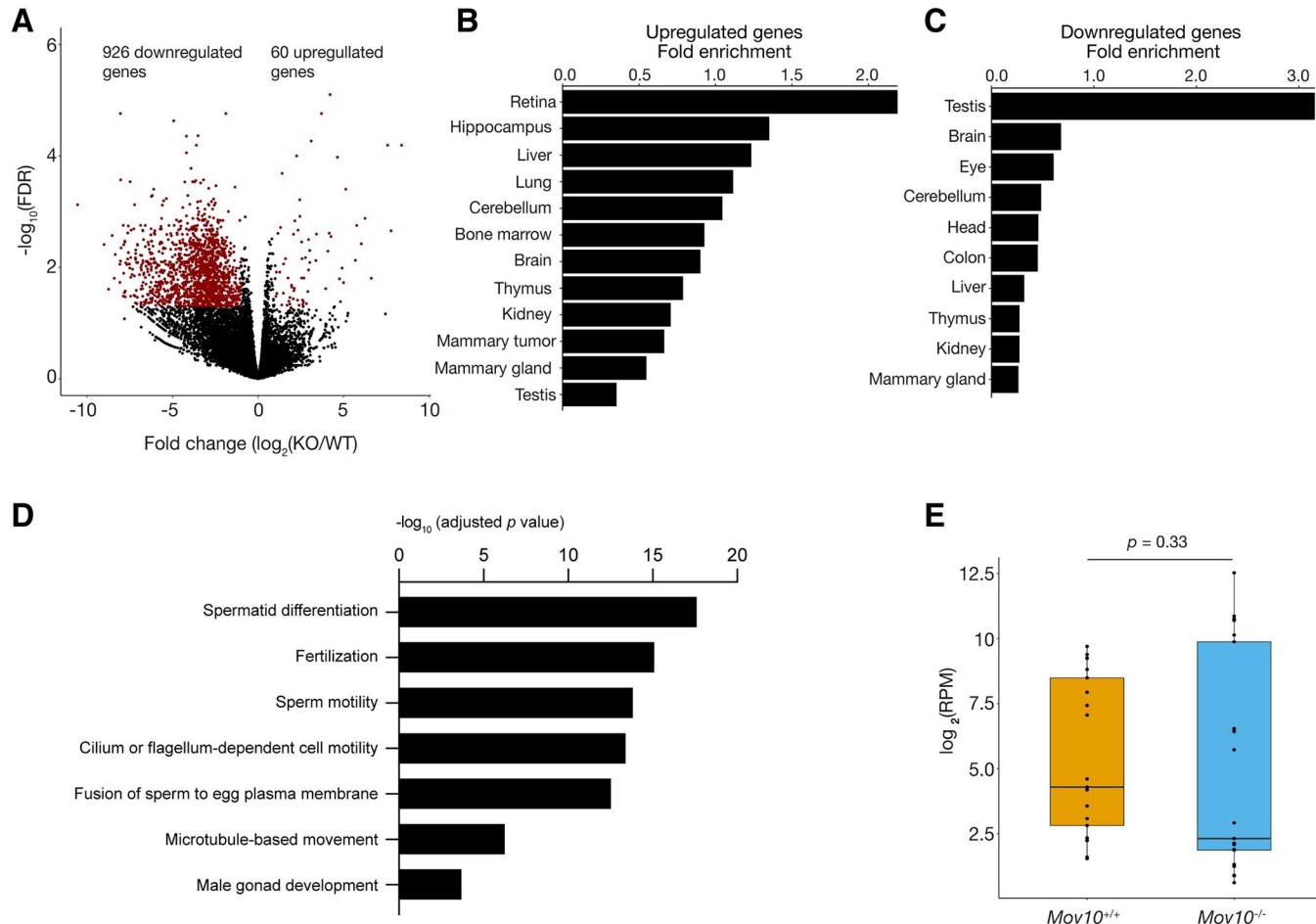

**Fig 3. Altered transcriptome in *Mov10*-deficient testes.** (A) Volcano plot of transcript levels of protein-coding genes between *Mov10*[+/+] and *Mov10*[-/-] testes at P21. Red dots indicate differentially expressed genes. (B) Relative expression of upregulated genes in *Mov10*[-/-] testes in different tissues. (C) Relative expression of downregulated genes in *Mov10*[-/-] testes in different tissues. (D) Enrichment of GO categories in down-regulated genes. (E) The expression of overall L1 elements in testes from P21 *Mov10*[+/+] and *Mov10*[-/-] mice. Each dot represents one L1 family.

should lack the intron and thus can be distinguished from the donor L1 transgene. The promoter of this L1[tg] transgene is methylated and thus repressed *in vivo* [55].

To trace new L1 insertions in the genome, we introduced this L1[tg] reporter transgene into our *Mov10*[-/-] mice. Intron-flanking PCR of genomic DNA identified new L1 copies in a subset of tissues from E18.5 embryos and 8-week-old *Mov10*[-/-] L1[tg/tg] mice (Fig 4B). Sequencing of the PCR bands confirmed that the new L1 insertions lacked the intron and thus originated from retrotransposition.

To compare the L1 insertion frequency in different tissues, we examined a total of 80 tissues from 10 *Mov10*[-/-] L1[tg/tg] embryos (5 males and 5 females; 8 tissues per embryo) at E18.5 and detected new L1 copies in all tissues, with higher insertion frequencies in spleen, ovary, liver, and kidney, but relatively lower insertion frequencies in lung, brain, and heart (Fig 4C). We then analyzed 224 tissues from 8-week-old *Mov10*[-/-] L1[tg/tg] mice (14 males and 14 females; 8 tissues per mouse) and found higher L1 insertion frequencies in spleen and kidney but lower frequencies in lung, brain, and heart (Fig 4C). MOV10 protein abundance was high in liver, kidney, and spleen (Fig 1A). These results showed that the new L1 insertion frequency in embryonic and adult *Mov10*[-/-] tissues correlated with the MOV10 protein abundance in wild

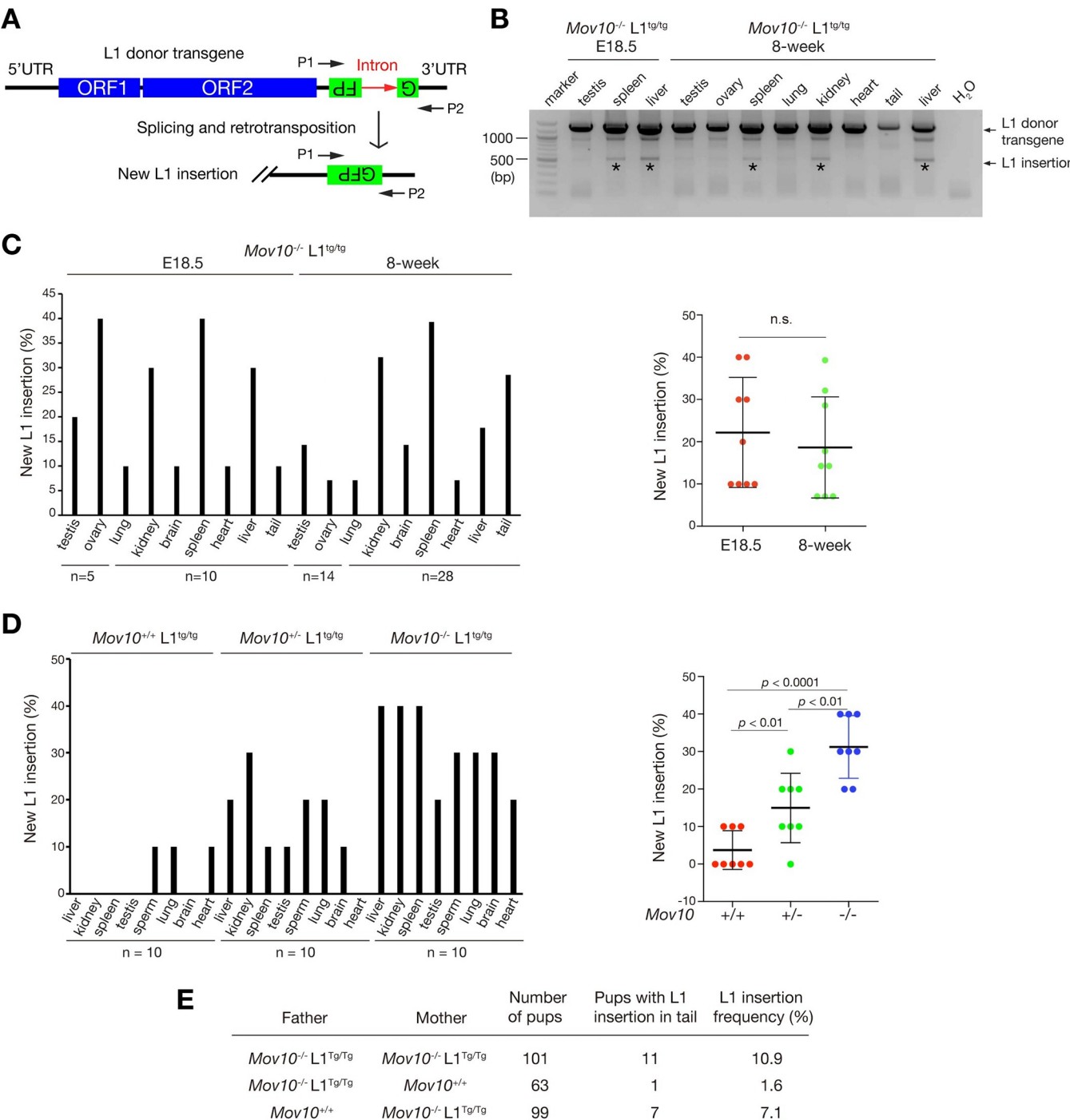

**Fig 4. MOV10 inhibits L1 retrotransposition *in vivo*.** (A) Schematic diagram of the single copy L1 reporter transgene (L1tg) as previously reported [55]. PCR primers P1 and P2 amplify both the donor transgene and new L1 insertion that lacks the intron. (B) The intron-flanking PCR detection of new L1 insertions in multiple tissues from E18.5 *Mov10*[-/-] L1tg/tg embryos and 8-week-old *Mov10*[-/-] L1tg/tg mice. The L1 donor transgene band (1.4 kb) and the new L1 insertion band (500 bp) are labeled. The black asterisk indicates the detected new L1 insertion band. (C) The percentage of the presence of new L1 insertions in tissues from E18.5 *Mov10*[-/-] L1tg/tg embryos and 8-week *Mov10*[-/-] L1tg/tg mice. Plot of L1 insertion frequency is shown on the right. Each point represents L1 insertion frequency (percentage) in one tissue. n.s.: non-significant. (D) The percentage of new L1 insertions in tissues from 8-week-old *Mov10*[+/+] L1tg/tg, *Mov10*[+/-] L1tg/tg, and *Mov10*[-/-] L1tg/tg mice. Plot of L1 insertion frequency is shown on the right. Each point represents L1 insertion frequency (percentage) in one tissue. Eight tissues (liver, kidney, spleen, testis, sperm, lung, brain, and heart) are included. Statistics, Student's *t*-Test. Numerical data are shown in S6 Table. (E) The L1 insertion frequency in tail genomic DNA of offspring from different crosses.

 

type tissues. In addition, the L1 insertion frequencies were similar between E18.5 tissues and adult tissues, indicating that the L1 insertion frequency in $Mov10^{-/-}$ tissues is likely to be age-independent (Fig 4C).

To further test whether MOV10 functions as a retrotransposition-restricting factor, we analyzed testis, epididymal sperm, and somatic tissues from $Mov10^{+/+}$ L1$^{tg/tg}$, $Mov10^{+/-}$ L1$^{tg/tg}$, and $Mov10^{-/-}$ L1$^{tg/tg}$ mice. We made several observations. First, new L1 insertions were rare but detectable in 10% of lung, heart, and sperm samples from the $Mov10^{+/+}$ L1$^{tg/tg}$ mice (Fig 4D). Second, compared with $Mov10^{+/+}$ L1$^{tg/tg}$ and $Mov10^{+/-}$ L1$^{tg/tg}$ mice, new L1 insertions in $Mov10^{-/-}$ L1$^{tg/tg}$ mice were increased, with the highest insertion frequencies in kidney, spleen, and liver (Fig 4D). Third, the new L1 insertion frequency was higher in tissues from $Mov10^{+/-}$ L1$^{tg/tg}$ mice than those from $Mov10^{+/+}$ L1$^{tg/tg}$ mice (Fig 4D), indicating that MOV10 inhibits L1 retrotransposition in a dosage-dependent manner. Fourth, L1 insertions were also detected in epididymal sperm, suggesting that these L1 insertions might be germline transmissible (Fig 4D). All these data support that MOV10 is a dosage-dependent inhibitor of L1 retrotransposition in both reproductive and somatic tissues in mice.

We analyzed new L1 insertions in tail genomic DNA of 101 offspring from the mating of $Mov10^{-/-}$ L1$^{tg/tg}$ males with $Mov10^{-/-}$ L1$^{tg/tg}$ females, and detected new L1 insertions in 11 pups (Fig 4E). We then examined the offspring from matings of $Mov10^{-/-}$ L1$^{tg/tg}$ males or females with wild type mice to distinguish male from female germline transmissions. Out of 63 offspring from the mating of $Mov10^{-/-}$ L1$^{tg/tg}$ males with $Mov10^{+/+}$ females, a new L1 insertion was detected in the tail of only one pup (Fig 4E). Out of 99 offspring from the mating of $Mov10^{-/-}$ L1$^{tg/tg}$ females with $Mov10^{+/+}$ males, new L1 insertions were detected in the tails of 7 pups (Fig 4E). However, we examined seven tissues (lung, kidney, brain, spleen, heart, liver, and testis or ovary) from 9 pups with L1 insertion-positive tails, and only one pup (from the $Mov10^{-/-}$ L1$^{tg/tg}$ intercross) had L1 insertions in all tissues examined. These results suggest that most L1 insertions detected in tail were not germline-transmitted but rather reflected new L1 insertions that occurred only in the tail.

## Impact of MOV10 deficiency on reproductive fitness over multiple generations

To test whether MOV10 deficiency affects reproductive fitness over multiple generations, we intercrossed $Mov10^{+/-}$ mice (G0) to obtain $Mov10^{-/-}$ mice (G1), which were then intercrossed to produce G2 $Mov10^{-/-}$ mice. Successive intercrosses were made until the sixth generation (G6) (Fig 5A). Three males from each generation were analyzed for testis weight, sperm count, testis histology, and litter size. The data from $Mov10^{+/-}$ mice (G0) served as the control. G1 $Mov10^{-/-}$ mice displayed similar parameters such as testis weight and sperm count as $Mov10^{+/-}$ mice (G0) (Fig 5B–5G). However, G2 $Mov10^{-/-}$ mice exhibited significant reductions in testis weight, sperm count, and an increased percentage of abnormal seminiferous tubules (Fig 5B–5D). The percentage of Sertoli cell-only tubules and litter size of G2 mice were similar with G0 mice (Fig 5E and 5F). G3 $Mov10^{-/-}$ mice showed further reduction in the testis weight and sperm count, and further increase in the percentage of abnormal tubules (Fig 5B–5D). The litter size of the mice decreased from 5.8 at G0 to 3.8 at G3, but the decrease did not reach statistical significance ($p = 0.10$, Fig 5F). These results showed that while the reproductive fitness of $Mov10^{-/-}$ mice decreased from G1 to G3, it did not continue to exacerbate in $Mov10^{-/-}$ mice beyond G3. Although the $Mov10^{-/-}$ mice from G4 to G6 still showed smaller testis, reduced sperm count, and defective spermatogenesis, the defects were less severe than in G3 mice (Fig 5B–5G). These results show that reproductive fitness of $Mov10^{-/-}$ mice worsens progressively until the third generation but recovers to a limited extent in later generations.

 

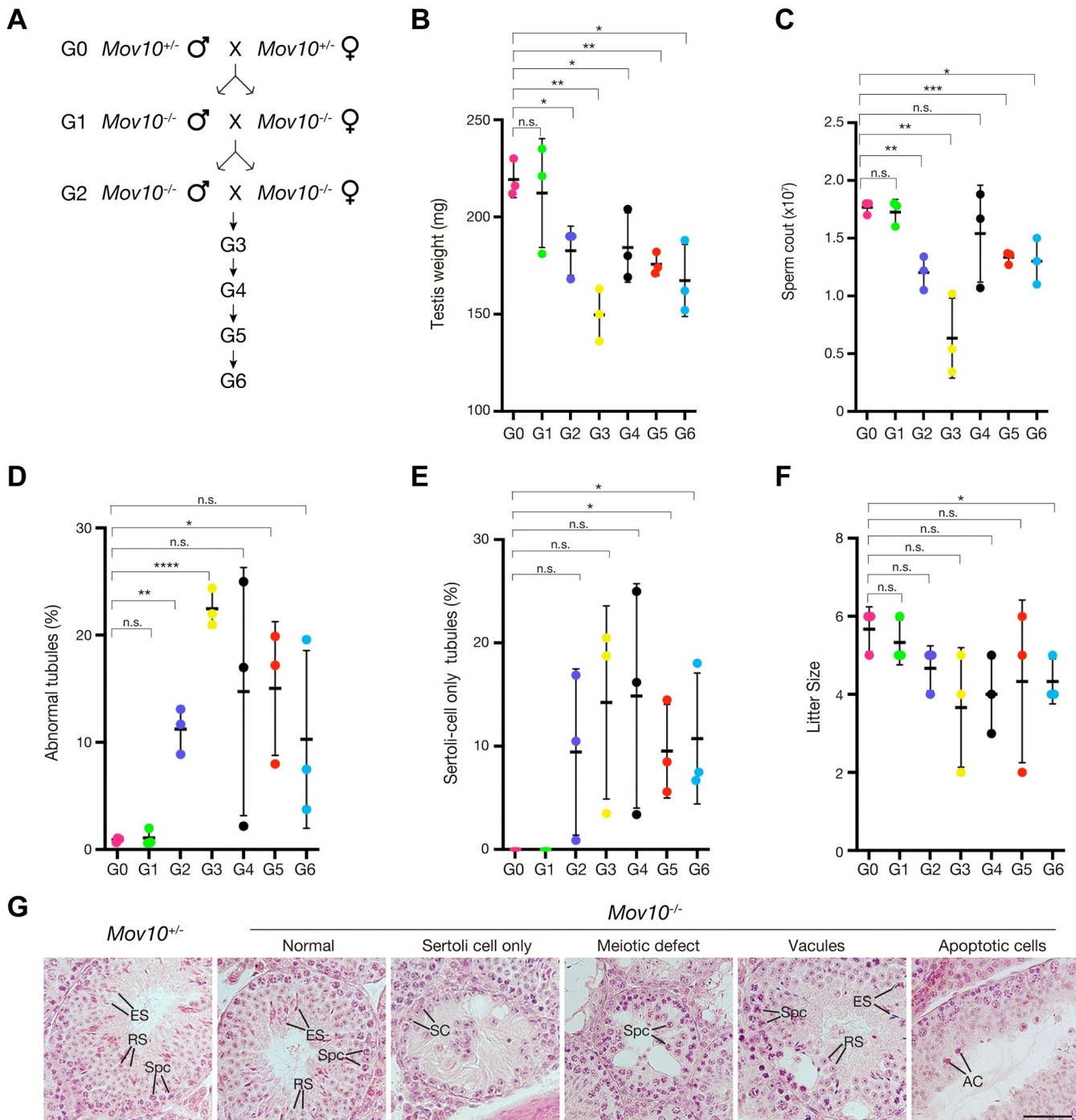

**Fig 5. Effect of MOV10 deficiency on the fertility parameters over successive generations.** (A) Breeding strategy for successive generations of *Mov10*<sup>-/-</sup> mice. (B-F) Analyses of 8-week-old *Mov10*<sup>-/-</sup> mice over six generations: testis weight (B), sperm count (C), percentage of abnormal tubules (D), percentage of Sertoli cell only tubules (E), and litter size (F). Analyses of G0 *Mov10*<sup>+/-</sup> mice were included for comparison. Statistics, Student's *t*-Test; n.s., non-significant; *, $p < 0.05$; **, $p < 0.01$; ***, $p < 0.001$; ****, $p < 0.0001$. Numerical data are shown in S6 Table. (G) Histology of testes from 8-week-old *Mov10*<sup>+/-</sup> and *Mov10*<sup>-/-</sup> mice. Abbreviations: SPC, spermatocyte; RS, round spermatids; ES, elongated spermatids; SC, Sertoli cells; AC, apoptotic cells. Scale bar, 50 μm.

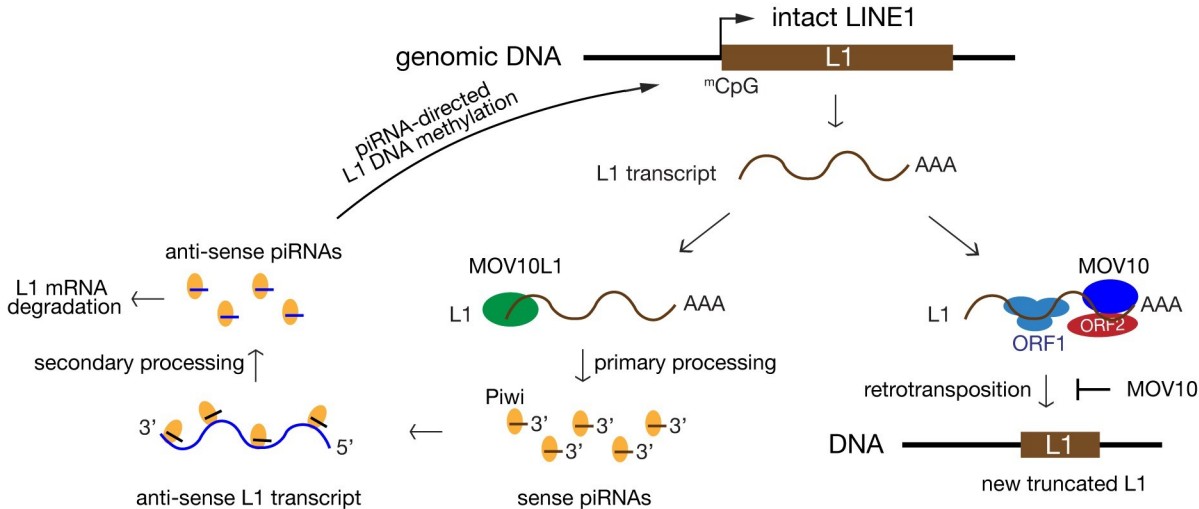

**Fig 6. A unifying model on the function of two related RNA helicases (MOV10 and MOV10L1) in the control of retrotransposons.** MOV10L1 is a master regulator of biogenesis of all piRNAs. MOV10 is the first *in vivo* host restriction factor of L1 retrotransposition. L1 transcript is normally present at a very low level, due to transcriptional silencing of L1 by methylation of the CpG dinucleotides in the L1 promoter. New L1 insertions are short (<1 kb) and often 5′ truncated due to incomplete reverse transcription.

## Discussion

While a multitude of transcriptional and post-transcriptional mechanisms exist to silence TEs, control of transposition *in vivo* remains poorly understood. Based on findings from this and previous studies, we propose a unifying model for the function of two homologous RNA helicases, MOV10 and MOV10L1, in suppression of retrotransposons in mouse. As both MOV10 and MOV10L1 are conserved in vertebrates including humans, we expect this model to be applicable to all vertebrates. Despite their sequence homology, our results show that MOV10 and MOV10L1 play distinct roles in control of retrotransposons (Fig 6). Previous studies revealed that MOV10L1 is essential for piRNA biogenesis in germ cells [27,28,30–32,56]. piRNAs orchestrate both post-transcriptional Piwi protein-dependent degradation of L1 transcripts and transcriptional silencing of active L1 transposons through de novo methylation of L1 promoters (Fig 6). In the absence of MOV10L1, L1 is highly upregulated in male germ cells [27] and a study using the L1^tg reporter assay has shown that loss of MOV10L1 leads to increased L1 retrotransposition in mouse germ cells [55].

In contrast to the germline-specific expression pattern of MOV10L1, MOV10 is expressed ubiquitously, and in cultured somatic cells, MOV10 strongly inhibits retrotransposition [42–45]. Here, we have demonstrated that MOV10 is an inhibitor of L1 retrotransposition in mouse (*in vivo*). MOV10 might regulate L1 retrotransposition through several mechanisms. First, MOV10 directly binds to a large number of transcripts including the L1 mRNA [37,57]. Our MOV10 CLIP-seq revealed that MOV10 preferentially binds to 3′ UTRs of transcripts in testis (Fig 2). MOV10 forms a complex with UPF1, an integral component of the NMD pathway, in HEK293T cells [37]. Our study further confirmed the interaction between MOV10 and UPF1 in mouse testis (S1 Fig). UPF1 binds to mRNA 3′ UTRs and is a highly processive helicase [49,50]. Therefore, MOV10 and UPF1 may target L1 mRNA for degradation through remodeling protein complexes on the RNA. In addition, MOV10 sequesters L1 ribonucleoprotein particles in the cytosolic aggregates [58]. However, loss of MOV10 did not change the

overall level of L1 transcripts in mouse testis. Second, MOV10 interacts with L1-encoded ORF2 and recombinant MOV10 blocks reverse transcription *in vitro* [48]. In addition, MOV10 interacts with TUT7 uridyltransferase. Uridylation of L1 3′ end inhibits initiation of reverse transcription [47]. Third, MOV10 interacts with RNASEH2 to degrade L1 mRNA in RNA-DNA hybrids. Knockdown of MOV10 or RNASEH2 in cells leads to accumulation of L1-specific RNA-DNA hybrids [46]. Lastly, since MOV10 is associated with L1 ribonucleic particles in cultured cells [34,35], it could inhibit L1 mobilization by interaction with additional proteins present in the particles. Thus, MOV10 likely inhibits LINE1 retrotransposition by multiple mechanisms.

In contrast with the presence of two related RNA helicases (MOV10 and MOV10L1) in vertebrates, *Drosophila* has only one homologous RNA helicase–Armitage, which functions in both the RNAi and piRNA pathways [59,60]. The RNA helicase activity is essential for retrotransposon control by both MOV10 and MOV10L1 [30,31,43]. In vertebrates, *Mov10* is likely to be the ancestral gene, because it is broadly expressed and inhibits retroviruses such as HIV-1, thus serving as an ancient innate anti-viral response. *Mov10l1* likely arose through gene duplication of *Mov10* and evolved to be germ cell-specific [26,61] to protect the genome integrity within the germ line from TEs throughout the vertebrate genomes. The functional specialization of MOV10L1 with the piRNA pathway, which most likely emerged later during evolution to deal with endogenous transposons, permitted generation of retrotransposon-specific piRNAs to mediate their targeted destruction, thus serving as a "de facto" adaptive anti-viral response to protect against retrotransposon mobilization. Thus, vertebrate MOV10 and MOV10L1 repress retrotransposons but by divergent mechanisms—inhibition of retrotransposition (MOV10) and production of piRNAs (MOV10L1) (Fig 6). However, we cannot rule out a possible overlap of function of MOV10 and MOV10L1 in piRNA biogenesis and L1 retrotransposition in germ cells.

While a previous study reported that inactivation of *Mov10* was completely embryonic lethal [48], in our study, the global knockout of *Mov10* caused only partial embryonic lethality and the surviving *Mov10*$^{-/-}$ mice were grossly normal and fertile. The difference in the viability of *Mov10*-deficient mice/embryos could potentially be attributed to genetic backgrounds. The genetic background of the mice in the Skariah study was C57BL/6 [48]. Our mice were on a hybrid genetic background (129 x C57BL/6 x Fvb) and survived, possibly due to hybrid vigor. Another possible explanation could lie in the difference of the knockout strategies. In the Skariah study, a gene trap *Mov10* mutant allele was used, whereas exons 8–14 of *Mov10* were deleted in our study (S3B Fig).

MOV10 is expressed in developing but not adult brain [48]. L1 is active in developing brain (mainly in hippocampus) and generates somatic mosaicism in neuronal progenitor cells [62]. We observed higher frequencies of new L1 insertions in all tissues including brain from *Mov10*$^{-/-}$ L1$^{tg/tg}$ mice. In addition, we observed elevated levels of new L1 insertions in tissues from *Mov10*$^{+/-}$ L1$^{tg/tg}$ mice in comparison with *Mov10*$^{+/+}$ L1$^{tg/tg}$ mice. Consistent with our results, Skariah *et al* reported that the L1 genomic content was higher in the *Mov10* heterozygous brain than the wild type [48]. These findings further support that MOV10 is a dosage-dependent inhibitor of L1 retrotransposition and suggests that the somatic mosaicism of L1 in neuronal progenitor cells may reflect cell-specific expression levels of MOV10. Such findings have broad implications, as it suggests that vertebrates including humans, with only one functional *Mov10* allele, may experience increased retrotransposition of TEs, potentially increasing the number of pathogenic TE insertions.

Despite increased retrotransposition, the surviving *Mov10*$^{-/-}$ mice were fertile. It is possible that most retrotransposition events occur in non-coding genomic regions or disrupt genes in a heterozygous manner. However, as these retrotransposition-mediated mutations accumulate

over multiple generations or become homozygous, defects are expected to develop in later generations. Indeed, *Mov10*$^{-/-}$ mice exhibited decreased testis weight, reduced sperm count, and increased defects in testicular histology over generations, with the most severe defects in the third generation. However, the fertility parameters did not continue to worsen beyond the third generation, possibly because gametes with a high load of insertions/mutations were selected against over multiple generations. In addition, MOV10 may regulate biological processes in addition to retrotransposition given that MOV10 binds to a large number of protein-coding transcripts. Therefore, the reproductive defects in *Mov10*$^{-/-}$ mice could be caused by or partially attributed to loss of L1-unrelated functions of MOV10.

MOV10 inhibits retrotransposition of not just L1 but also SINE and IAP retrotransposons in cell culture-based assays [42–45], raising the possibility that it may act as a host restriction factor for all active TEs *in vivo*. The mouse genome harbors ~3000 intact L1 elements. The mouse genome also contains the intracisternal A-particle (IAP) element, an active rodent LTR retrotransposon that is absent in human. SINEs are non-coding elements and hitchhike on L1-encoded proteins for retrotransposition [63,64]. In this study, we tracked the generation of new L1 insertions in tissues from *Mov10*$^{+/-}$ and *Mov10*$^{-/-}$ mice using the L1 reporter transgene. Given the abundance of endogenous retrotransposons, the increase in new insertions derived from the single-copy L1 reporter in *Mov10*$^{+/-}$ and *Mov10*$^{-/-}$ mice could be only the tip of an iceberg in the mobilization landscape of endogenous retrotransposons (L1, SINE, and IAP) in *Mov10* mutant mice.

## Materials and methods

### Ethics statement

Mice were maintained and used for experimentation according to the guidelines of the Institutional Animal Care and Use Committees of the University of Pennsylvania under protocol# 806616.

### Targeted inactivation of the *Mov10* gene

In the *Mov10* targeting construct, a 2.3-kb genomic DNA segment harboring exons 8–14 was flanked by loxP sites (S3B Fig). The two homologous arms (2.3 kb and 1.95 kb) were amplified from a *Mov10*-containing BAC clone (RP24-503L4) by PCR with high-fidelity DNA polymerase. The HyTK selection cassette was cloned before the right arm. V6.5 ES cells were electroporated with the linearized targeting construct, cultured in the presence of hygromycin B (120 μg/ml, Invitrogen), and screened by long-distance PCR for homologously targeted *Mov10*$^{3lox}$ clones. Two *Mov10*$^{3lox}$ ES cell lines were electroporated with the pOG231 plasmid that expresses the Cre recombinase. ES cells were subjected to negative selection with gancyclovir (2 μM; Sigma) for removal of the HyTK cassette. ES cell colonies were screened by PCR. Recombination between the immediate HyTK-flanking loxP sites resulted in the *Mov10*$^{fl}$ allele (S3B Fig). Two *Mov10*$^{fl}$ ES cell lines were injected into B6C3F1 (Taconic) blastocysts that were subsequently transferred to the uteri of pseudopregnant ICR females. The *Mov10*$^{fl}$ allele was transmitted through the germline from chimeric males. *Actb*-Cre mice were used to generate *Mov10* mutant allele [53]. The *Ddx4*-Cre (240 bp) was genotyped with primers CACGTG-CAGCCGTTTAAGCCGCGT and TTCCCATTCTAAACAACACCCTGAA. The *Amh*-Cre transgene (305 bp) was genotyped with primers GCATTACCGGTCGATGCAACGAGTG and GAACGCTAGAGCCTGTTTTGCACGTTC. Intercrosses of *Mov10*$^{+/-}$ mice were used to generate *Mov10*$^{-/-}$ mice. The *Mov10* wild type allele (325 bp) was assayed with primers ATCGCCACTAGCCCTGAAGCAT and GCCGCATAGAAACTTAGATCCATCC. The

*Mov10*⁻ (knockout) allele (315 bp) was assayed by PCR with primers GCGGTTGTTACAAGA AGGAGTTCTCA and GCCGCATAGAAACT-TAGATCCATCC.

## Genotyping of L1 5′ UTR-ORFeus transgene mice

The L1 5′ UTR-ORFeus transgene (L1$^{tg}$) was produced before [55]. The L1 transgene maps to the first intron of the *Tnr1* gene. The wild type (*Tnr1*) allele (356 bp) was genotyped with ACT-GAGTGACCTCGGGTATATTTC and CTGCTGAGCTGTTGTAACTCCTT. The L1 transgene allele (156 bp) was assayed with CGGGCCATTTACCGTAAGTTATGT and CTGCATTCTAGTTGTGGTTTGTCCA. The L1 transgene was introduced into *Mov10*-deficient background by breeding. The parental L1$^{tg}$ allele (1401 bp) and new L1$^{tg}$-derived insertions (500 bp) were genotyped with intron-flanking primers ACCCAACACCCGTGCGTT TTATT and TGGAGTACAACTACAACAGCCACAACGTCT (P1 and P2, Fig 4A).

## Histological and immunofluorescence analyses

For histological analysis, testes and ovaries were fixed in Bouin's solution at room temperature overnight, embedded with paraffin, and then sectioned at 5 μm. Sections were stained with hematoxylin and eosin. As for immunofluorescence analysis, testes were fixed in 4% paraformaldehyde (in 1×PBS) for 6 hours at 4˚C, dehydrated in 30% sucrose (in 1×PBS) overnight and sectioned at 5 μm. The primary and secondary antibodies used for immunofluorescence analyses were listed in S3 Table.

## Immunoprecipitation, mass spectrometry, and Western blotting analyses

For immunoprecipitation (IP), 100 mg P21 testes were lysed in 1 ml RIPA buffer (10 mM Tris, pH 8.0, 140 mM NaCl, 1% Trion X-100, 0.1% sodium deoxycholate, 0.1% SDS, 1 mM EDTA) supplemented with 1 mM PMSF. Cell lysates were centrifuged by 16,000 g for 30 min at 4˚C, and 1.5% of the supernatant was set aside as input. The remaining lysates were pre-cleared with 30 μl protein G Dynabeads (Thermo Fisher Scientific) for two hours, and then incubated with 3 μg MOV10 antibody (A301-571A, Bethyl lab) or UPF1 antibody (A301-902A, Bethyl lab) or normal rabbit IgG (2729, Cell Signaling) at 4˚C for 1 hour. The lysates were then incubated with 30 μl protein G Dynabeads overnight. The immunoprecipitated complexes were washed with the RIPA buffer three times and boiled in 30 μl 2× SDS-PAGE loading buffer at 95˚C for 10 minutes. 25 μl of the supernatant was resolved by SDS-PAGE. For MOV10 mass spectrometry, the gel was stained with Coomassie Blue dye, and the bands that were present in the MOV10 lane but absent in the IgG lane were sent for mass spectrometry at Wistar Institute Proteomics Core. For western blot analysis, the resolved proteins were transferred onto a nitrocellulose membrane using iBlot (Invitrogen) and immunoblotted with primary and secondary antibodies (S3 Table).

## MOV10 HITS-CLIP

MOV10 HITS-CLIP was performed as described previously [65]. Testes from P21 mice were collected, de-tunicated, dissociated by mild pipetting in ice-cold HBSS, and followed by UV crosslinking three times at 400 mJ/cm², with 30-second intervals for cooling. Testicular cells were pelleted at 1200 g for 10 min at 4˚C, washed with 1×PBS, then the cell pellet was snap-frozen in liquid nitrogen and kept at -80˚C if not used immediately. UV light-treated cells (from two testes) were lysed in 700 μl of 1×RIPA buffer (10 mM Tris-HCl, pH 8.0, 140 mM NaCl, 1% Trion X-100, 0.1% sodium deoxycholate, 0.1% SDS, 1 mM EDTA) with 1 mM PMSF at 4˚C

for 1 hour. After that, lysates were treated with 10 μl DNase (Promega) and 2 μl RNase T1 at 37°C for 5 min. The lysates were centrifuged at 90,000g for 30 min at 4°C.

For each immunoprecipitation, 3 μg of rabbit anti-MOV10 polyclonal antibody (A301-571A, Bethyl lab) was bound on protein A Dynabeads in the antibody-binding buffer (0.1 M $Na_3PO_4$, pH 8, 0.1% IGEPAL CA-630, 5% Glycerol) at 4°C for 3 hours, and then antibody-bound beads were washed three times with 1×PBS. Antibody-bound beads were incubated with lysates at 4°C for 3 hours. Ligation of the $^{32}$P labeled RL3 RNA adapter was described before [65]. Immunoprecipitation beads were eluted for 10 min at 70°C using 30 μl 2×SDS sample buffer. The eluted samples were separated by 10% precast gels (Biorad, 4561033). Cross-linked RNA–protein complexes were transferred onto the nitrocellulose membrane (Invitrogen, LC2001), and then the membrane was exposed to film overnight. Membrane regions containing the main radioactive signal and up to 15 kDa higher were cut. RNA extraction, 5′ linker ligation (RL5), reverse transcription and two rounds of PCR were performed as described previously [65]. The sequences of the primers for the first PCR (DP3 and DP5) and the second PCR (DSFP3 and DSFP5) were available in S4 Table. The DNA products were resolved on 3% agarose gels and extracted with QIAquick gel extraction kit and submitted for deep sequencing. Four libraries with different indexes (S4 Table) were sequenced on the Illumina HiSeq 2500 platform at 100 cycles. MOV10 HITS-CLIP data were analyzed as previously described [30]. The read density of each CLIP-binding transcript was normalized to reads per million reads (RPM) with the formula: RPM = (number of reads mapped to transcripts/total number of mapped reads) × 1,000,000. The MOV10 HITS-CLIP-seq data are available under the GEO accession no: GSE217336.

The expression of MOV10-bound transcripts in mouse somatic tissues was obtained by re-analyzing the published RNA-seq data from these 3-week-old mouse tissues: liver, bone marrow, bone, retina, kidney, prefrontal cortex, and brain stem (S5 Table).

## RNA-seq and data analysis

Total RNA was isolated from 6 pairs of P21 mouse testes (3 pairs of wild type and 3 pairs of $Mov10^{-/-}$) using TRIzol reagents (Thermo Fisher Scientific). 1 μg of total RNA from each sample was used to generate RNA-seq libraries using TruSeq Stranded mRNA Library Preparation Kit Set A (Cat. No. RS-122-2101, Illumina) according to the manufacturer's instruction. The quality of libraries was evaluated using the Agilent 4200 TapeStation (Agilent Technologies). All 6 individual libraries were pooled in equal amount and sequenced with the HiSeq 4000 platform (Illumina). The RNA-seq data are available under the GEO accession no: GSE217336.

Adapter sequences were trimmed and the low-quality reads were removed. The clean reads were mapped to the mouse genome (mm10) using STAR with default parameters. The number of reads mapped to each gene was counted by htseq-count (http://www-huber. embl.de/users/anders/HTSeq/) based on the annotation from ENSEMBL (http://uswest. ensembl.org/) mouse gene annotation v99. The expression of transposable elements was analyzed using TEtranscripts [66]. Identification of differentially expressed genes was performed by edgeR. Differential expression was defined as a fold change greater than 2 and false discovery rate (FDR) < 0.05. FDR was calculated based on Benjamini and Hochberg multiple testing correction.

## Statistics

Statistical analysis was performed with Student's *t*-test, if not otherwise described.

## Supporting information

**S1 Fig. MOV10 forms a complex with UPF1 in mouse testis.** (A) Identification of MOV10-associated proteins in lysates from postnatal day 20 (P20) mouse testes by immuno-precipitation and mass spectrometry. The gel was stained with Coomassie Blue dye. The two bands indicated by vertical lines in the MOV10 immunoprecipitation lane were subjected to protein identification by mass spectrometry. (B) Co-immunoprecipitation analysis of MOV10 and UPF1 in P20 *Mov10*$^{+/+}$ and *Mov10*$^{-/-}$ testes. IP was performed with anti-MOV10 antibody. (C) Reciprocal co-immunoprecipitation analysis of MOV10 and UPF1 in P20 wild type testes. IP was performed with anti-UPF1 antibody.
(TIF)

**S2 Fig. Expression and subcellular localization of MOV10 in mouse testis and ovary.** (A) Western blot analysis of MOV10 in developing mouse testes. The timing of the first appear-ance of spermatogonia, preleptotene spermatocytes, pachytene spermatocytes, round sperma-tids, and elongated spermatids in developing testes is shown. ACTB serves as a loading control. (B) Immunofluorescence of MOV10 in frozen sections of mouse testes at different ages and 8-week-old ovary. The expression levels of MOV10 are depicted in different colors at the bottom diagram. The stage of seminiferous tubules in 8-week-old testis is shown in roman numerals. GC, gonocytes; Sg, spermatogonia; SC, Sertoli cells; Lep, leptotene; Zyg, zygotene; Pa, pachytene; RS, round spermatids; ES, elongated spermatids. Scale bar, 50 μm.
(TIF)

**S3 Fig. The domain structure of mouse MOV10, targeted inactivation of the *Mov10* gene, and *Mov10* mouse mutants.** (A) Schematic diagram of the mouse MOV10 RNA helicase domains. The two different RNA helicase domains are color coded. (B) Targeted inactivation of the *Mov10* gene. Targeting vector, floxed conditional allele, and knockout allele are shown. The *Mov10* gene has 21 exons based on the cDNA under accession number NM_001163440.1. Deletion of exons 8–14 encoding aa 455–807 removes the first RNA helicase domain (in green) and results in a frame shift in the resulting transcript. The protein regions encoded by colored exons match those in panel A. (C) Immunofluorescence of MOV10 in frozen sections of testes from 8-week-old *Mov10*$^{+/+}$, *Mov10*$^{fl/fl}$ *Amh-Cre* (Sertoli cell-specific conditional knockout), *Mov10*$^{fl/-}$ *Ddx4-Cre* (Germ cell-specific conditional knockout), and *Mov10*$^{-/-}$ (global knockout) mice. The meshwork-like green fluorescence pattern in *Mov10*$^{fl/-}$ *Ddx4-Cre* testis section is due to the expression of MOV10 in the cytoplasm of Sertoli cells, but is absent in *Mov10*$^{fl/fl}$ *Amh-Cre* testis section as expected. Sg, spermatogonia; PreL, preleptotene sper-matocytes; Lep, leptotene spermatocyte; Sertoli, Sertoli cells. Scale bar, 50 μm.
(TIF)

**S4 Fig. Histology of testes and ovaries from 8-week-old wild type and *Mov10*$^{-/-}$ mice.** Arrows in the ovaries indicate follicles. Abbreviations: Pa, pachytene spermatocytes; RS, round spermatids; ES, elongated spermatids. Scale bars, 50 μm.
(TIF)

**S1 Table. MOV10 CLIP-seq data.**
(XLSX)

**S2 Table. List of DE genes in *Mov10* KO testes and GO analysis.**
(XLSX)

**S3 Table. Primary and secondary antibodies.**
(DOCX)

**S4 Table. Primers used in MOV10 HITS-CLIP.**
(DOCX)

**S5 Table. RNA-seq datasets from postnatal day 21 (P21) mouse tissues used in Fig 2E.**
(DOCX)

**S6 Table. Numerical data underlying graphs in Figs 4 and 5.**
(XLSX)

## Acknowledgments

We thank Wenfeng An for the L1<sup>tg</sup> mice, Hsin-Yao Tang and Thomas Beer for mass spectrometry, Jonathan Schug for Next-gen sequencing, Shantan Reddy, Aoife Roche, and Frederic Bushman for advice on the retrotransposition analysis. We thank Frederic Bushman and Leslie King for critical reading of the manuscript.

## Author Contributions

**Conceptualization:** Yongjuan Guan, P. Jeremy Wang.

**Data curation:** Hongyan Gao.

**Formal analysis:** Yongjuan Guan, Hongyan Gao, Anastassios Vourekas, Panagiotis Alexiou, Manolis Maragkakis, Zissimos Mourelatos, Guanxiang Liang, P. Jeremy Wang.

**Funding acquisition:** Zissimos Mourelatos, P. Jeremy Wang.

**Investigation:** Yongjuan Guan, Anastassios Vourekas, Zhenlong Kang.

**Methodology:** Yongjuan Guan, N. Adrian Leu, Anastassios Vourekas.

**Resources:** P. Jeremy Wang.

**Supervision:** P. Jeremy Wang.

**Validation:** Yongjuan Guan, P. Jeremy Wang.

**Visualization:** Yongjuan Guan, P. Jeremy Wang.

**Writing – original draft:** Yongjuan Guan, P. Jeremy Wang.

**Writing – review & editing:** Yongjuan Guan, Anastassios Vourekas, Zissimos Mourelatos, Guanxiang Liang, P. Jeremy Wang.

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
