## [Decision Letter · Decision Letter 0]

11 Jan 2023

Dear Dr. Wang,

Thank you very much for submitting your Research Article entitled 'The MOV10 RNA helicase is a dosage-dependent host restriction factor for LINE1 retrotransposition in mice' to PLOS Genetics.

The manuscript was fully evaluated at the editorial level and by independent peer reviewers. The reviewers appreciated the attention to an important problem, but raised some substantial concerns about the current manuscript. Based on the reviews, we will not be able to accept this version of the manuscript, but we would be willing to review a much-revised version. We cannot, of course, promise publication at that time.

If you decide to revise the manuscript for further consideration at PLOS Genetics, please aim to resubmit within the next 60 days, unless it will take extra time to address the concerns of the reviewers, in which case we would appreciate an expected resubmission date by email to plosgenetics@plos.org.

Please do not hesitate to contact us if you have any concerns or questions.

Yours sincerely,

Alex Bortvin, M.D., Ph.D.

Guest Editor

PLOS Genetics

Gregory Barsh

Editor-in-Chief

PLOS Genetics

Reviewer's Responses to Questions

**Comments to the Authors:**

Reviewer #1: Comments

The authors investigate the role of an RNA helicase MOV10 in mice and describe its role in retrotransposon repression. The authors demonstrate a role that is distinctly different from that reported previously by the authors for its paralog MOV10L1. Loss of Mov10 causes low-penetrant lethality, but those who make it beyond the embryonic stages appear normal. They identify MOV10-bound transcripts to show that the protein binds the 3’UTR of several mRNAs and a select set of L1 retrotransposons, and these L1 elements are upregulated in the KO germline. Using a L1 transposition reporter, they identify increased transposition events in the KO mouse tissues, that is also seen at reduced levels in Mov10 HET animals, indicating a dosage effect. These transposition events result in increased impact on gene expression changes over successive generations, perhaps due to accumulation of mutations. Taken together, this nicely executed study demonstrates a role for MOV10 in retrotransposon control using the mouse model and quantifies it using a L1 transposition reporter. A role for regulation of non-germline transcripts (mainly soma-expressed) is also proposed. I support its publication after the minor concerns are addressed.

Minor.

1. The role of UPF1 in L1 inhibition by MOV10 should be clearly discussed. UPF1 is also shown to bind to mRNA 3’UTRs (Hurt et al., 2013) and is a highly processive helicase (Fiorini et al., 2015) that may remodel protein complexes on the RNA. Since MOV10 and UPF1 can bind RNA independently, is there any overlap of RNAs bound by the two proteins? I don’t know if dataset of CLIPseq of UPF1 in mouse testis is available? Please compare and discuss, if such a dataset is available.

2. Line 154: The statement should refer to the figure where this data is shown, if it exists? I assume it’s unpublished data or part of Figure S3C?

3. Line 182-193: It is not clear if the statements convey the intended meaning. This is what I understand, MOV10 binds RNAs which are degraded. In the Mov10 mutant testis, these RNAs are upregulated (Fig 3D). Please check.

4. Line 198: Again, there is a contradiction in the two statements on L1 and SINE elements being regulated by MOV10. Please check.

5. In the results section, please clearly state where the conditional KO is used vs the clean KO.

Reviewer #2: 

In this manuscript, the authors used Mov10 knockout and conditional knockout mouse models and L1 retrotransposition reporter mice to evaluate a role for Mov10 in inhibiting L1 retrotransposition in vivo. They reveal that unlike Mov10l1, an Mov10 family member that regulates L1 through the piRNA pathway, Mov10 suppresses L1 transposition in various mouse tissues. This is a significant advance in understanding the distinctive mechanisms for the Mov10 family RNA helicases in L1 regulation.

Comments:

It would be helpful to show GFP protein expression data in Mov10-/- L1tg/tg tissues such as GFP immunofluorescence, and/or LINE1 immunofluorescence in Mov10-/- L1 tissue. This will further strengthen the conclusion.

Line 284-301: the authors indicate that MOV10 is a “potent” inhibitor of L1 retrotransposition in mouse (in vivo) and MOV10 regulates L1 retrotransposition through several mechanisms. How to define “potent”, this should be backed by data.

Line 260: “G2 Mov10-/- mice exhibited significant reductions in testis weight, sperm count, and the percentage of abnormal seminiferous tubules.” Should be an increase of percentage of abnormal seminiferous tubules.

Is reproductive fitness defect described in Fig. 5 the result of misregulation of L1 retrotransposition? How would the authors explain L1 unrelated function of Mov10 in germ cell regulation?

The model in Fig. 6: can the author exclude any overlapping function between Mov10 and Mov10l1 in piRNA biogenesis or in L1 retrotransposition?

Reviewer #3:

The paper would be of interest to the fields of mobile elements, virology, and development. Based on the title and abstract, the main conclusions the authors want to put forth are that MOV10 KO leads to L1 up regulation in testes, and subsequently L1 retrotransposition. I think this is a good first effort however the depth of analysis, conclusions, and data presentation needs to be strengthened. There does seem to be a clear trend that there is more retrotransposition in MOV10-/- mice, but given that MOV10 seems to bind to and affect so many transcripts, this modest retrotransposition increase could very well be an indirect effect. Other descriptions of the MOV10 phenotype are given (incomplete embryonic lethality, reduced reproductive fitness over successive generations), but it is unclear whether these are related to the L1 phenotype. Retrotransposition is occurring at quite low frequencies in wt, so the slight bump in frequency may not have any obvious biological consequence in the experiments shown.

Points of concern:

1) There is already a published mouse KO of MOV10 (Skariah et al 2017), but the authors inexplicably do not mention this until the the end of their manuscript, halfway through the discussion. Plausible explanations for the differences in embryonic lethality are given, such as potentially different background, but then the authors claim that Skariah et al do not report the genetic background of their mouse which is incorrect (the background is C57BL/ 6 as described in additional file 3). Although the Skariah et al paper has some dubious conclusions (e.g. the “doubling of genomic L1 in MOV10 heterozygotes” is highly suspect), the existence of this literature should be acknowledged and differences or problems with the paper mentioned upfront. The focus of Skariah et al was different from this manuscript so it should not negatively affect this manuscript.

2) Line 115 why is b-actin used as a loading control when it is barely expressed and heart/ skeletal muscle? How can you make conclusions about MOV10 expression in heart/muscle based on this figure?

3) Figure S1. There appear to be many bands specific to the MOV10 IP. What was the rationale for only choosing 4 bands to identify? What are the other bands besides UPF1? Why highlight other bands that were identified without saying what they are?

4) Figure 2 - Although the authors describe the general patterns on MOV10 RNA binding (preferentially binds to 3’UTR, binds to low abundance RNAs), I’m puzzled as to why they did not go into more depth on the names of the RNAs that were bound. Do they fall into any particular functional categories? Can they explain the MOV10 phenotype? Are retroelement sequences overrepresented in the MOV10 strong binders? It seems like a lot of work to produce this data, only to gloss over the details.

5) I am confused by the interpretation of Figure 3. If the authors are proposing that MOV10 binds to and down regulates RNAs, why does the knockout of MOV10 lead to massive down regulation of the transcriptome? Figure 2D seems to show many, many RNAs that are low expression, strong MOV10 binding, but then knockout of MOV10 only leads to 60 RNAs being unregulated while >900 are down regulated? Figure 3D picks 20 transcripts that are strongly bound to MOV10 and and have increased expression in the MOV10 KO, but this seems to be cherry picking. Aren’t these a small minority of the MOV10 bound RNAs? What if the authors analyzed the entire MOV10 bound population?

6) The analysis of L1 transcription is problematic. There are many fragments of L1 in the genome that are incorporated into the transcript of other genes. Some of the most

abundant transcripts containing L1 are not bona fide L1 transcription, but fragments of L1 in the 3’ UTR of an mRNA. The analysis pipeline used to measure L1 family transcripts does not take this locus specific information into account and thus we do not know if these increases represent an increase of authentic L1 transcription. I don’t believe any solid conclusions can be made about L1 expression from this analysis, and the same could be possibly be said about the other elements.

7) Figure 4 - there seems to be redundancy in the subfigures. Why not combine the 8 week KO samples from figure 4C and 4D? Are they essentially the same? It is hard to make conclusions about which tissues are higher or lower, as the percentages for same tissues and genotype are very different in some cases (e.g. adult KO lung is <10% in figure 4C but presumably the same experiment adult KO lung is 30% in figure 4D).

Minor comments:

- Where appropriate, please cite primary literature. For example, line 75 Dewannieux and Heidmann (2003 and 2005) are more appropriate citations than references 12 and 13.

- Line 81 Carmell et all (2006) and some others should be added as primary references.

- Line 97 Moldavan and Moran (2015) is another proteomic study that should be cited.

- line 730 (Figure 2D legend) what is 3 weeks and what is adult in this subfigure?

- Lines 241-247. The experiment does not test germline transmission, so why even start this

paragraph stating that you were trying to test germline transmission? I would just state the results. If this data (figure 4F) was broken down into the different pup genotypes, there might also be some interesting information there.

**Have all data underlying the figures and results presented in the manuscript been provided?**

Reviewer #1: Yes

Reviewer #2: Yes

Reviewer #3: **No: **I could not access the sequencing data (did not get a token)

PLOS authors have the option to publish the peer review history of their article (what does this mean?). If published, this will include your full peer review and any attached files.

Reviewer #1: No

Reviewer #2: No

Reviewer #3: No

---

## [Decision Letter · Decision Letter 1]

28 Mar 2023

Dear Dr. Wang,

Thank you very much for submitting the revised version of your Research Article entitled 'The MOV10 RNA helicase is a dosage-dependent host restriction factor for LINE1 retrotransposition in mice' to PLOS Genetics.

The manuscript was fully evaluated at the editorial level and by independent peer reviewers. The reviewers are largely satisfied with changes that you made to the paper and data presentation. Nevertheless, we would like you address the remaining minor critique by Reviewer 3 before moving forward with your study. If Editors find your response satisfactory, we will be able to make the final decision without the need for another round of reviews. We ask you to modify the manuscript according to the review recommendations. Your revisions should address the specific points made by each reviewer. In addition we ask that you to provide a detailed list of your responses to the review comments and a description of the changes you have made in the manuscript.

Yours sincerely,

Alex Bortvin, M.D., Ph.D.

Guest Editor

PLOS Genetics

Gregory Barsh

Editor-in-Chief

PLOS Genetics

Reviewer's Responses to Questions

**Comments to the Authors:**

Reviewer #1: I commend the authors for the revision. I find the revised version to my satisfaction. It will contribute to our evolving understanding of the protein.

Reviewer #2: The authors have addressed all my questions.

Reviewer #3: The manuscript is greatly improved and I only have one point of concern regarding figures 2C/2D. It would be useful to include in the materials how this read density was normalized. For example, how does the read density of L1_MM shown in figure 2D compare to the other transcripts displayed in figure 2C? It is hard to compare without units on the y-axis. How would the data of figure 2D look if superimposed on figure 2C? Also, the text says that MOV10 CLIP-seq density is enriching in the L1 3' UTR, but the pattern is strikingly different between figures 2C and 2D, with most transcripts having a huge bump is read density throughout the 3'UTR compared to coding regions. The L1 actually has very similar levels of read density of the 3' UTR and coding region, with the exception of the terminus of the 3'UTR (which may be the polyA tail?). Does figure 2D take into account that the L1 3' end is more abundant in the genome and thus in total RNA?

I do not need to see the manuscript again assuming this concern is adequately answered.

**Have all data underlying the figures and results presented in the manuscript been provided?**

Reviewer #1: Yes

Reviewer #2: Yes

Reviewer #3: Yes

PLOS authors have the option to publish the peer review history of their article (what does this mean?). If published, this will include your full peer review and any attached files.

---

## [Editor Report · Decision Letter 2]

14 Apr 2023

Dear Dr. Wang,

We are pleased to inform you that your manuscript entitled "The MOV10 RNA helicase is a dosage-dependent host restriction factor for LINE1 retrotransposition in mice" has been editorially accepted for publication in PLOS Genetics. Congratulations!

Yours sincerely,

Alex Bortvin, M.D., Ph.D.

Guest Editor

PLOS Genetics

Gregory Barsh

Editor-in-Chief

PLOS Genetics

Comments from the reviewers (if applicable):

**Data Deposition**

http://datadryad.org/submit?journalID=pgenetics&manu=PGENETICS-D-22-01422R2

**Press Queries**

---

## [Editor Report · Acceptance letter]

27 Apr 2023

PGENETICS-D-22-01422R2 

The MOV10 RNA helicase is a dosage-dependent host restriction factor for LINE1 retrotransposition in mice 

Dear Dr Wang, 

We are pleased to inform you that your manuscript entitled "The MOV10 RNA helicase is a dosage-dependent host restriction factor for LINE1 retrotransposition in mice" has been formally accepted for publication in PLOS Genetics! Your manuscript is now with our production department and you will be notified of the publication date in due course.

With kind regards,

Anita Estes

PLOS Genetics

On behalf of:
